# OpenReview forum: "ToolTweak: An Attack on Tool Selection in LLM-based Agents"
_ICLR.cc/2026/Conference — Submitted to ICLR 2026_

### Official Review · Reviewer_jvww · 2025-10-19

**Soundness:** 2
**Presentation:** 3
**Contribution:** 2
**Rating:** 2
**Confidence:** 4

**Summary:**

The paper introduces ToolTweak, a gradient-free adversarial attack that iteratively rewrites a tool’s name and description using an attacker LLM to increase its selection rate by LLM-based agents. The attack assumes the adversarial tool provider can modify their own tool’s metadata and observe usage statistics. The authors demonstrate that ToolTweak significantly increases selection rates across multiple models and also evaluate simple paraphrasing defenses.

**Strengths:**

The paper is well written and easy to read.

The authors clearly present the setup, problem formulation, and threat model.

**Weaknesses:**

1. Editing tool descriptions to increase selection rates has already been explored in prior work, including [1] and [2]. This makes the novelty and contribution of this paper incremental. Additionally, the authors fail to cite and discuss [2], which presents a very similar black-box attack.


2. The assumptions made in ToolTweak are strong: the attack requires access to usage statistics and full visibility into the entire tool database. This level of access is unrealistic for many practical deployment settings. Even under this setup, as shown in Table 1, the attack underperforms compared to simple manual suffix addition.

3. The defense strategies evaluated are relatively simple and resemble standard jailbreak mitigation techniques. A deeper exploration of defenses specifically tailored to agentic LLM tool selection would be more meaningful and impactful.

4. In the main results (Section 4.3), the attacker and victim models are the same. This setup risks overfitting to model-specific biases and undermines generalizability.

[1] Faghih, Kazem, et al. "Gaming Tool Preferences in Agentic LLMs."

[2] Chen, Liuji, et al. "Select Me! When You Need a Tool: A Black-box Text Attack on Tool Selection."

**Questions:**

See above.

---

### Official Review · Reviewer_QcA4 · 2025-10-27

**Soundness:** 2
**Presentation:** 4
**Contribution:** 2
**Rating:** 4
**Confidence:** 4

**Summary:**

This paper investigates vulnerabilities in tool selection within large language model agents. It introduces ToolTweak, a black box optimization procedure that edits tool names and descriptions to increase the chance that a target tool is chosen. Experiments show strong effects across several models, with selection rates rising from about twenty percent to more than eighty percent. The authors also evaluate simple mitigation methods such as paraphrasing and perplexity filtering.

**Strengths:**

The paper highlights a realistic security and fairness issue in tool based ecosystems that has not been thoroughly explored before.
It provides clear empirical results with consistent quantitative gains across different models.
The work connects technical findings to broader implications for fairness and competition, which adds practical value.

**Weaknesses:**

The technical novelty of the attack is limited. The idea of iteratively optimizing textual metadata to maximize the probability of a desired token output has already been widely used in earlier adversarial text generation research. The contribution mainly lies in applying such optimization to the tool selection setting.

The optimization process is relatively simple and lacks deeper analysis of efficiency or convergence. It is closer to an empirical heuristic than a new algorithmic framework.

The baseline comparisons are narrow. The experiments mainly contrast ToolTweak with simple suffix modifications but do not include stronger adversarial optimization baselines such as gradient based or reinforcement learning based methods.

The evaluation of defenses is rather basic. The two proposed methods, paraphrasing and perplexity filtering, are intuitive but limited. It would be more convincing to include comparisons with other feasible defenses such as metadata normalization, usage based anomaly detection, or classifier based filtering that detects subjective or promotional language in tool descriptions.

**Questions:**

See weaknesses.

---

### Official Review · Reviewer_sLJw · 2025-10-30

**Soundness:** 2
**Presentation:** 3
**Contribution:** 2
**Rating:** 2
**Confidence:** 4

**Summary:**

This paper proposes an attack method targeting the LLM-based proxy tool selection process, ToolTweak. By iteratively optimizing the tool name and description, it significantly increases the probability of the target tool being selected. Experiments show that this method can increase the tool selection rate from approximately 20% to a maximum of 81% on multiple models, and demonstrates strong cross-model migration. The paper also evaluated two defense methods (interpretation and PERPLEXITY filtering), and analyzed the potential impact of attacks on the fairness and security of the tool ecosystem.

**Strengths:**

+ The issue is of practical significance: With the widespread use of tool invocation in LLM proxies, bias and attacks in the tool selection process are indeed a security and fairness issue worthy of attention.
+ The experimental design is systematic and comprehensive: multiple open-source and closed-source models were evaluated, and the migration of attacks, defense effects, and influencing factors (such as tool sequence, parameter complexity, etc.) were analyzed.
+ Defense methods are evaluated: two defense methods are explored, and their effectiveness was analyzed, providing a reference for subsequent research.

**Weaknesses:**

- The threat model is overly assumptions: attackers are assumed to have access to all the tool information in the entire tool library, which may not be realistic in actual attack scenarios and limits the practicality and universality of the method.
- The innovation of the method is limited: Tool selection attacks have been involved in existing studies (such as《From Allies to Adversaries: Manipulating LLM Tool-Calling through Adversarial Injection》). Essentially, the method proposed in this paper is still metadata manipulation based on iterative optimization, and the technical method is not novel enough.
- The implementation details are not described clearly: Although there are appendices to supplement, the specific mechanism by which LLMS generate "bias" names and descriptions during the attack process, as well as the convergence conditions for iterative optimization and other key details, are still not transparent enough, affecting the feasibility of reproduction.

**Questions:**

- In the actual tool platform, is it possible for attackers to obtain all the tool information? Is there an attack scheme under a weaker assumption?

- Compared with the existing tool selection attack methods, what is the innovation of ToolTweak? Are there any significant advantages in terms of attack efficiency, concealment, and mobility?

---

### Official Review · Reviewer_aZub · 2025-11-01

**Soundness:** 2
**Presentation:** 4
**Contribution:** 3
**Rating:** 4
**Confidence:** 4

**Summary:**

This paper introduces ToolTweak, a gradient-free adversarial attack that manipulates the names and descriptions of tools to bias tool selection in LLM-based agents. The attack is automatic and transferable, significantly increasing a targeted tool’s selection rate (e.g., from ~20% to over 80%) across various agents and tasks. The authors also analyze its broader impact on fairness and security in tool ecosystems and propose a paraphrasing defense, showing that the vulnerability remains a persistent challenge.

**Strengths:**

1. This paper is identifying a newly occured research problem with function calling.

2. The draft is super clear and concise to read.

**Weaknesses:**

See questions.

**Questions:**

1. Thought this paper is proposing a new research question, I am not convinced on this point. Because tool selection a ranking task, any tool providers can optimize their tools through black box method (acturally they should do). Will the problem still exist if all tools are optimized using the same TOOLTWEAK method?

2. I think a more in-depth discussion on "how the distributional shifts in tool usage will impacting the product" is necessary. I currently did not see how large impact it will cause as we should have tool providers' competition to ensure tool improvement.

3. Do authors think a uniform distribution is the optimal one and should not be tweaked?

---

### Meta-Review · Area_Chair_XMcb · 2025-12-10

**Summary:**

The paper proposes ToolTweak, a black-box, gradient-free adversarial attack designed to manipulate Large Language Model (LLM) agents into selecting specific tools by optimizing tool names and descriptions. The authors demonstrate that this method can increase selection rates significantly (from ~20% to ~80%) and show transferability across different models. They also evaluate defenses such as paraphrasing and perplexity filtering.

While the reviewers acknowledged the practical significance of the problem and the clarity of the presentation, the consensus is leaning towards Rejection. The primary concerns stem from limited technical novelty compared to existing adversarial text generation methods, a threat model that makes overly strong assumptions about the attacker's knowledge of the tool database, and insufficient experiments.

**Reviewer Concerns:**

Since the authors did not provide a response, the reviewers' concerns were not well addressed.

**Reviewer Scores:**

Given the lack of a rebuttal to address the critical issues regarding novelty and the threat model, it is unlikely the reviewers would have raised their scores.

---

### Decision · Program_Chairs · 2026-01-26

Reject